# Clustering of Sedentary Behaviours, Physical Activity, and Energy-Dense Food Intake in Six-Year-Old Children: Associations with Family Socioeconomic Status

**DOI:** 10.3390/nu12061722

**Published:** 2020-06-09

**Authors:** Junwen Yang-Huang, Amy van Grieken, Lu Wang, Wilma Jansen, Hein Raat

**Affiliations:** 1The Generation R Study Group, Erasmus Medical Center, 3015 GD Rotterdam, The Netherlands; j.yang@erasmusmc.nl; 2Department of Public Health, Erasmus Medical Center, 3015 GD Rotterdam, The Netherlands; a.vangrieken@erasmusmc.nl (A.v.G.); Lu.Wang@tufts.edu (L.W.); w.jansen@Rotterdam.nl (W.J.); 3Friedman School of Nutrition Science and Policy, Tufts University, Boston, MA 02111, USA; 4Municipality of Rotterdam, 3011 AD Rotterdam, The Netherlands

**Keywords:** social inequalities, lifestyle behaviour cluster, overweight/obesity

## Abstract

This study examined the clustering of lifestyle behaviours in children aged six years from a prospective cohort study in the Netherlands. Additionally, we analysed the associations between socioeconomic status and the lifestyle behaviour clusters that we identified. Data of 4059 children from the Generation R Study were analysed. Socioeconomic status was measured by maternal educational level and net household income. Lifestyle behaviours including screen time, physical activity, calorie-rich snack consumption and sugar-sweetened beverages consumption were measured via a parental questionnaire. Hierarchical and non-hierarchical cluster analyses were applied. The associations between socioeconomic status and lifestyle behaviour clusters were assessed using logistic regression models. Three lifestyle clusters were identified: “relatively healthy lifestyle” cluster (*n* = 1444), “high screen time and physically inactive” cluster (*n* = 1217), and “physically active, high snacks and sugary drinks” cluster (*n* = 1398). Children from high educated mothers or high-income households were more likely to be allocated to the “relatively healthy lifestyle” cluster, while children from low educated mothers or from low-income households were more likely to be allocated in the “high screen time and physically inactive” cluster. Intervention development and prevention strategies may use this information to further target programs promoting healthy behaviours of children and their families.

## 1. Introduction

Childhood obesity is a major public health problem in most developed and developing countries [1]. In 2014, the average percentage of overweight and obese children was 19% in Europe [2]. The primary causes of overweight and obesity in children can be traced to various lifestyle behaviours related to imbalance between calorie intake and energy expenditure [1].

Research on co-occurrence or clustering of energy-related lifestyle behaviours, such as dietary behaviours, sedentary behaviours, and physical activity in children has increased [3,4,5,6,7]. It has been shown that healthy and unhealthy behaviours co-occur in children in complex ways [8]. Evaluating the synergetic effect instead of the isolated effects of lifestyle behaviours will help intervention development to further target lifestyle behaviours simultaneously [9]. Furthermore, studies have shown that socioeconomic status (SES) is associated with certain lifestyle behaviour clusters [10,11,12,13]. For example, Leech et al. found that a higher proportion of children aged 10–12 years with mothers having low educational level tended to be in the “energy dense food/drink consumers who watch TV” cluster [10]. Ottevaere et al. reported that adolescents aged 12.5–17.5 years with higher educated parents were more likely to be in the “healthy” cluster and the “healthy eating, low physical activity and low sedentary behaviour” cluster [13]. Research on the associations between SES and lifestyle behaviour clusters may be helpful to identify subgroups at increased risk in developing overweight and obesity.

A systematic review pointed out that few studies have examined the clustering of lifestyle behaviours among children younger than nine [4,7,14,15]. Identifying the clustering of lifestyle behaviours in school-aged children is important, since screen behaviour, physical activity, and dietary behaviours are established in early childhood and can be tracked into later life [16,17]. Among school-aged children, besides their own preferences, parents play an important role in the development of children’s lifestyle behaviours through their parental attitudes, parenting practices, financial capabilities, and personal lifestyle behaviours. In the studies focusing on socioeconomic inequalities in clustering of lifestyle behaviours, parental educational level was the most common measure of SES [4]. Other indicators of SES have been researched sparsely [18,19]. Studying a variety of SES indicators may provide a complete overview of the impact of socioeconomic status on the clustering of child lifestyle behaviours [20].

This study firstly examined the co-occurring patterns of lifestyle behaviours, including screen time, physical activity, calorie-rich snack consumption, and sugar-sweetened beverages consumption, in children aged six years from a prospective cohort study in the Netherlands. Secondly, we analysed the associations between SES, measured by both maternal educational level and net household income, and the lifestyle behaviour patterns that we identified.

## 2. Materials and Methods

### 2.1. Study Design

The study was embedded in the Generation R Study. The Generation R Study is a prospective population-based birth cohort in Rotterdam, The Netherlands. The cohort includes 9778 mothers and their children who were born between 1 April 2002 and 31 January 2006 [21]. Consent for follow-up was available for 8305 children at aged 6 years. Children with information on lifestyle behaviours (i.e., screen time, physical activity, calorie-rich snack, and sugar-sweetened beverages) available were included in the study (*n* = 4516). In total, 12 children did not have data on maternal educational level and net household income; these cases were excluded. Second (*n* = 293) and third children (*n* = 6) of the same mother were excluded for analyses to avoid clustering. Univariate outliers (i.e., screen time > 6 h/day, physical activity > 6 h/day, calorie-rich snack > 4 portion/day, sugar-sweetened beverages > 7 portion/day) were removed, leaving a study population of 4059 participants. The study was approved by the Medical Ethics Committee of the Erasmus University Medical Centre (MEC 217.595/2002/202). Written informed consent was obtained from all participants.

### 2.2. Socioeconomic Status

Maternal educational level was obtained via questionnaire when the child was 6 years old using the Dutch Standard Classification of Education. Four education levels were categorized: low (no education, primary school, lower vocational training, intermediate general school, or four years or less general secondary school), mid-low (more than four years general secondary school, intermediate vocational training, or first year of higher vocational training), mid-high (higher vocational training), and high (university or PhD degree) [22]. Net household income was obtained by questionnaire when the child was 6 years old and categorized as low (<€2000/month), middle (€2000–€3200/month), or high (>€3200/month).

### 2.3. Lifestyle Behaviours

Children’s lifestyle behaviours, including total screen time, physical activity, calorie-rich snack, and sugar-sweetened beverages, were measured using a parental questionnaire when the child was age 6.

#### 2.3.1. Screen Time

Parents reported children’s time spent on television viewing and computer playing respectively. For television viewing time, parents were asked to report the average number of days per week (0–5 days) and per weekend (0–2 days) their child spent watching television, videos, or DVDs. On the days that their child spent watching television, videos or DVDs, parents reported the average number of hours in the morning, afternoon, and evening after dinner per weekday/weekend day. The average time children spent on television viewing per day was calculated by the following formula: [weekdays × (hours in the morning + hours in the afternoon + hours in the evening after dinner) + weekend days × (hours in the morning + hours in the afternoon + hours in the evening after dinner)]/7. The same set of questions was used to assess children’s time spent behind a computer, which included game computers such as a PlayStation, Gameboy and Nintendo. The average computer time per day of the child was calculated according to the same formula as for television time. Total screen time per day was calculated by adding up children’s television time and computer time.

#### 2.3.2. Physical Activity

Parents reported children’s sports participation and outdoor play respectively. For sports participation, parents were asked to name the sport that their children took part in. Frequency (i.e., number of times per week) and duration (i.e., average hours for each training session or match) were reported. Response categories for frequency ranged from ‘1 time per week’ to ‘more than 3 times per week’. Response categories for duration included: ‘less than 30 min’, ‘30 to 60 min’, and ‘more than 1 h’. The average time the child spent on sport per day was calculated using the following formula: times per week × average hours each session/7. School physical educational lessons and swimming lessons were assessed separately and were not included in the assessment of sports participation.

Parents reported the frequency (i.e., number of days) and duration (i.e., average hours in the morning, afternoon, or evening after dinner) of children’s outdoor play for weekdays and weekend days separately. Response categories for duration included: ‘never’, ‘less than 30 min’, ‘30–60 min’, ‘1–2 h’, ‘2–3 h’, and ‘3–4 h’. The average outdoor play time per day was calculated using the following formula: [weekdays × (hours in the morning + hours in the afternoon + hours in the evening after dinner) + weekend days × (hours in the morning + hours in the afternoon + hours in the evening after dinner)]/7. Physical activity time per day was calculated by adding up children’s sports participation and outdoor play.

#### 2.3.3. Calorie-Rich Snack

Consumption of calorie-rich snacks was assessed by the following question for weekdays and weekend days separately: How often, on average, does your child eat a calorie-rich snack? The following definition was provided to parents: a calorie-rich snack is something that is eaten between the three main meals, such as chips, nuts, chocolate bars, cookies, or ice cream. Response categories for this question included: ‘never or less than once per day’, ‘once per day’, ‘2–3 times per day’, ‘4–6 times per day’, and ‘7 or more times per day’. The middle number of portions of each category (e.g., 5 portions for 4–6 times per day) was used to estimate the average consumption of calorie-rich snacks. The number of snacks on weekdays and weekend days was summed up and then divided by seven days to calculate the average total calorie-rich snack consumption per day.

#### 2.3.4. Sugar-Sweetened Beverages

Consumption of sugar-sweetened beverages was assessed using the following question for weekdays and weekend days separately: On average, how many glasses/packages of sugar-sweetened beverages does your child drink? Parents received the following definition of sugar-sweetened beverages: sugar-sweetened beverages are those beverages containing a great deal of (added) sugar, including soft drinks, fruit juices, lemonade, and sweetened milk products (e.g., chocolate milk). Response categories ranged from ‘none or less than 1’ to ‘7 or more’ (8 categories in total). The number of sugar-sweetened beverages on weekdays and weekend days were summed and then divided by seven days to calculate the average total sweet beverage consumption per day.

### 2.4. Potential Confounders

Based on the literature, several characteristics were considered potential confounders in the analyses: child sex (boy/girl), age (years), ethnic background, and child weight status [4,23]. Information on child ethnic background (western, non-western) was based on the parents’ country of birth, which was obtained by questionnaire when the child was 6 years old. If one of the parents was born outside the Netherlands, this country of birth determined the ethnic background of the child. If both parents were born outside the Netherlands, the country of birth of the mother determined the ethnic background of the child [24]. Height and weight were measured in lightweight clothes and without shoes, at the Generation R research center in the Erasmus Medical Center, Sophia’s Children’s Hospital. Body mass index (BMI) was calculated using the formula: weight (kilograms) divided by height (meters) squared. Children were categorized into overweight (including obesity) and normal weight according to international age- and sex-specific BMI cut-off points [25].

### 2.5. Statistical Analyses

To identify clusters of children with similar lifestyle co-occurring patterns, a combination of hierarchical and non-hierarchical cluster analyses were used [23]. Log transformation was applied to the four lifestyle behaviour variables because of positive skewedness. Z-scores of the log-transformed variables were calculated to standardize the variables before cluster analysis. First, a hierarchical cluster analysis was applied using Ward’s method based on Euclidean distance [26]. At this stage, several possible cluster solutions with the number of clusters ranged from 3 to 6 were generated. Second, a non-hierarchical k-means cluster analysis was performed using the initial cluster centres generated from the hierarchical cluster analysis. Third, to test the stability of the generated cluster solutions, 50% of the study population was randomly selected and the clustering procedure was repeated. The agreement of the cluster assignment between the main study population and the randomly selected sample was assessed with Cohen’s kappa (ĸ) [27].

Chi-square tests were performed to investigate the differences with regard to the cluster distribution by child characteristics and family SES. In each cluster, odds ratios for different SES indicators (high maternal educational level and high net household income as reference group) were calculated using logistic regression. Potential confounders (child sex, age, ethnic background, and child weight status) were included into the models. Bonferroni correction was applied for multiple testing [*p* = 0.05/(cluster number × number of SES indicators)]. Interaction effects between child sex and SES indicators were assessed in the logistic regression models. No statistically significant interaction effects were found (*p* < 0.05). Multiple imputation procedures were performed to impute missing data in the determinants and confounders (ranging from 0% to 9.3%, Table 1) using a fully conditional specified model. Five imputed datasets were generated, taking into account all the variables included in this study. Pooled estimates were used to report odds ratios (ORs) and their 95% confidence intervals (CIs). Statistical analyses were performed using IBM SPSS Statistics for Windows, version 24.0. Armonk, NY, USA: IBM Corp.

### 2.6. Non-Response Analyses

Children with missing data on at least one life style behaviour (*n* = 3789) were compared with children without missing data (*n* = 4516) using Chi-square tests. Data were more often missing for children from mothers with a low educational level, a low household income, or from non-western ethnic background (all *p* < 0.05). No statistically significant differences were found between boys and girls (*p* = 0.64).

## 3. Results

Table 1 shows the characteristics of children and their mothers. The mean age of the children was 6.0 (SD 0.4) years. Approximately 30% of the mothers had a high educational level. More than half of children (54.2%) lived in a high income household. Around three quarters (74.9%) of the children had a western ethnic background.

### 3.1. Description of the Clusters

Based on the four lifestyle behaviours, cluster analyses turned out a three-cluster solution (κ agreement = 0.964) as the most adequate and stable representation. Figure 1 presents the three clusters derived from the cluster analysis. Cluster 1 was labelled “relatively healthy lifestyle”, and it was characterized by z-scores < 0 for total screen time, calorie-rich snack and sugar-sweetened beverages consumption, and relatively high in physical activity level (z-score = 0.21). Cluster 2 was labelled “high screen time and physically inactive”, and it was characterized by high total screen time level (z-score = 0.33) and low physical activity level (z-score = −0.90). Cluster 3 was labelled “physically active, high snacks and sugar-sweetened beverages”, and it was characterized by high physical activity level (z-score = 0.56), high calorie-rich snack consumption (z-score = 0.64), and high sugar-sweetened beverages consumption (z-score = 0.57). The means and standard deviations of lifestyle behaviours for each cluster are presented in Table 2.

### 3.2. Cluster Distribution according to Child Characteristics and Socioeconomic Status

Table 3 presents the cluster distribution according to child characteristics and SES indicators. Boys were most allocated in the “physically active, high snacks and sugar-sweetened beverages” cluster (54.1%) (*p* < 0.001). The “high screen time and physically inactive” cluster showed the highest proportion of children being overweight/obese (15.9%) (*p* < 0.001). Significant differences in clusters were found by both maternal educational level and net household income (*p* < 0.001). The “relatively healthy lifestyle” cluster showed the highest proportion of children from mothers with a high educational level (40.1%) and children from families with a high-income household (62.9%) (*p* < 0.001). The “high screen time and physically inactive” cluster showed the highest proportion of children from mothers with a mid-low educational level (34.9%) and children from families with a low-income household (27.0%) (*p* < 0.001).

### 3.3. Associations of SES Indicators with the Cluster Distribution

Table 4 presents the results from multinomial logistic regression models for the associations between SES indicators and the lifestyle behaviour clusters among children. An adjusted *p*-value [*p* = 0.05/(3 × 2) = 0/008] was applied since a three-cluster solution was identified. Compared to children of mothers with a high educational level, children of mothers with a low educational level had an OR of 0.28 (95% CI: 0.21, 0.37) to be allocated in the “relatively healthy lifestyle” cluster. On the contrary, compared to children of mothers with a high educational level, children of mothers with a low educational level had an OR of 1.45 (95% CI: 1.13, 1.86) to be allocated in the “high screen time and physically inactive” cluster and an OR of 2.28 (95% CI: 1.79, 2.90) to be in the “physically active, high snacks and sugary drinks” cluster.

Compared to children from high-income households, children from low-income households had an OR of 0.59 (95% CI: 0.48, 0.74) to be allocated in the “relatively healthy lifestyle” cluster and an OR of 1.57 (95% CI: 1.27, 1.94) for the “high screen time and physically inactive” cluster.

## 4. Discussion

In this study, we explored clusters of lifestyle behaviours in a large sample of six-year-old children in the Netherlands. Healthy or unhealthy levels of lifestyle behaviours co-occurred in some groups. Three clusters were observed: “relatively healthy lifestyle”, “high screen time and physically inactive”, and “physically active, high snacks and sugary drinks”. Children from high educated mothers or high-income households were more likely to be allocated to the “relatively healthy lifestyle” cluster, while children of low educated mothers or from low-income households were more likely to be allocated to the “high screen time and physically inactive” cluster.

More than one third of the children in our study sample were allocated to the “relatively healthy lifestyle” cluster. Children in this cluster, on average, achieved more than 1 h/day of physical activity [28]. Total screen time use was, on average, below the recommended 2 h/day [29]. On average, children in this cluster consumed one portion of calorie-rich snack and one portion of sugar-sweetened beverage per day, which was the lowest amount in all three clusters observed. Similar types of clusters defined by low sedentary behaviour and low snack and beverage consumption have been observed by other studies among children of different ages as well [9,11,30]. For example, Bel-Serrat et al. reported that a “low beverage consumption and low sedentary” cluster was observed among children aged three to six years living in eight European countries [30]. Another study conducted by Bel-Serrat et al. identified a “low beverage intake, low sedentary, and physically active” cluster among children aged six to nine years living in 17 European countries [9]. Matias et al. observed a “health-promoting sedentary behaviour and diet” cluster in a sample of over 100,000 children aged 14 years in Brazil [11]. In addition, we found that the “relatively healthy lifestyle” was more likely to be observed in children of mothers with a high educational level or children from a high-income household. Parents with high SES may be more inclined to use and adhere to information concerning healthy lifestyles and be more competent to offer healthy choices to their younger children compared to low SES parents [13].

Children in the “high screen time and physically inactive” cluster have the lowest level of physical activity of the three observed clusters. Although the average screen time use was just about 2 h/day, it was the highest level of the three clusters. Such displacement between sedentary behaviour and physical activity has been reported in previous studies [10,23]. A systematic review showed that among several studies, many clusters were defined by high levels of sedentary behaviour [4]. Regardless of being combined with other healthy/unhealthy lifestyle behaviours or not, clusters defined by high levels of sedentary behaviour were associated with an increased risk of overweight/obesity [9].

Consistent with previous studies [4,10], we found that the “high screen time and physically inactive” cluster was more likely to be observed in children of mothers with a low educational level or children from a low-income household. A study conducted among children from seven European countries aged 10–12 years old also found that children with low educated parents were more likely to be allocated to a low activity/sedentary cluster or sedentary and sugared drinks cluster [23]. These results demonstrated that children from low SES backgrounds tend to be more prevalent in clusters combining multiple unhealthy lifestyles. In our study, sports participation was assessed and included as one form of physical activity. For low SES parents of young children, the lack of resources to sign their children up for a sports activity (e.g., football, judo, gymnastics, jazz, ballet, tennis, etc.) might play an important role. This may explain the social inequality we found in the “high screen time and physically inactive” cluster. In addition, our results showed that a higher proportion of boys were in the “high screen time and physically inactive” cluster, unlike in a systematic review which reported that girls were more likely to be in the low physical activity clusters [4]. Meanwhile, our results also showed that boys were more often in the “physically active, high snacks and sugary drinks” cluster. One possible explanation is that the gender differences in physical activity may link to the child’s age. Previous studies were mostly conducted in older children or adolescents. The gender differences in physical activity were larger in adolescents than in younger children [31]. Furthermore, boys and girls have been shown to have different sedentary behaviour [32]. We observed similar results that boys spent more time watching television and playing computer games than girls, which may explain the higher proportion of boys in the “high screen time and physically inactive” cluster. Future studies may use the information from this study to develop and evaluate programs that use clusters of lifestyle behaviours in order to provide support to children and their families.

In this study, high physical activity level was observed co-existing with high calorie-rich snacks and sugary drinks consumption. To the best of our knowledge, this is the first study to identify a “physically active, high snacks and sugary drinks” cluster in children at this young age group. The co-occurrence of high physical activity and high calorie-rich snacks and drinks consumption is consistent with a review in adults that reported exercise-induced increase in energy intake is typically compensated for by energy-dense food and drinks [33]. Consumption of calorie-rich snacks and sugary drinks may attenuate the beneficial effects of physical activity on skeletal mass [34] and the maintenance of body weight [33]. We also found that children of mothers with low educational level had higher odds of being allocated in the “physically active, high snacks and sugary drinks” cluster, but household income was not associated with being allocated to this cluster. It has been suggested that parental educational level has an independent association with child lifestyle behaviours [35,36]. Educational level could reflect the level of parental knowledge on healthy lifestyle behaviours and therewith impact the availability and opportunity for children to engage in healthy lifestyle behaviours [37]. This is especially relevant for young school-aged children, who still spend most of their time at home and are less affected by peer behaviour, as compared to older school-aged children. The co-occurrence of high physical activity and high calorie-rich snacks and drinks consumption exists in adults [38], and this could impact parental practices related to healthy lifestyle behaviours. Further research is warranted to confirm the findings of this cluster in relatively young children. In addition, examining how the clustering of lifestyle behaviours progresses over time from a younger age can provide more insight into the changes of children’s lifestyle behaviours.

### Methodological Considerations

The main strength of our study is the availability of information on lifestyle behaviours in a large sample of school-aged children. Some limitations should be considered. First, net household income was measured via a self-reported questionnaire, and therefore social desirability cannot be excluded. Around 5% of the data was missing. It cannot be ascertained whether an individual tends to over-report or under-report the household income. Second, all child lifestyle behaviours included in the current study were self-reported by the parents, which may have led to bias. Parents’ reports of physical activity may have been underestimated as outdoor play and sports participation may also occur in settings outside the home environment (e.g., school and after-school care). Although detailed frequency and duration/portion of each behaviour was measured in the questionnaire, and separately for weekdays and weekends, other measures of children’s lifestyle behaviours such as diaries or the use of activity trackers can provide additional information in future research. Research is needed to examine the possibilities of using the identification of clusters of lifestyle behaviour in youth health care practice. Third, only children with complete data on four lifestyle behaviours were included in the study population. Particular characteristics of the excluded participants may bias the cluster distribution. Finally, the causality for the associations of SES with lifestyle behaviour clusters cannot be established from observational studies only. Future studies are needed to establish causal relationships.

## 5. Conclusions

Our study showed three clusters of co-occurring patterns with regard to screen time, physical activity, and energy-dense food intake among children aged six years in the Netherlands. Only one third of the children were allocated to the relatively healthy cluster. Other clusters identified showed healthy or unhealthy trends in co-occurrence with lifestyle behaviours. A higher maternal educational level was associated with higher odds for the child to be allocated to the relatively healthy lifestyle behaviour cluster. Children from low-income households were more likely to be allocated to one of the relatively unhealthy lifestyle behaviour clusters, compared to children from a high-income household. Intervention development and prevention strategies may use this information to further target programs promoting healthy behaviours of children and their families.

## Figures and Tables

**Figure 1 nutrients-12-01722-f001:**
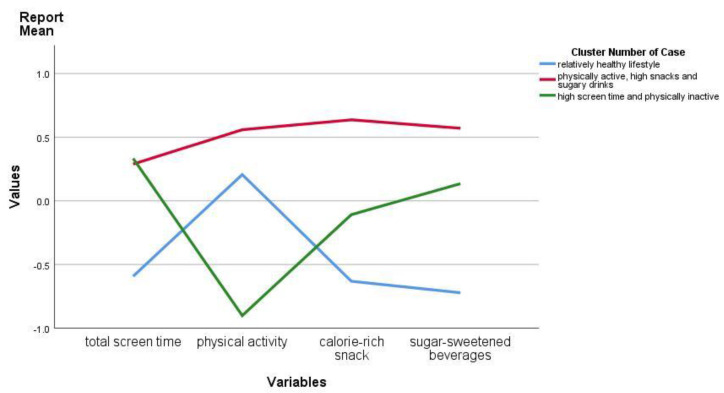
Z-scores of clusters on child lifestyle behaviours in participants from the Generation R Study.

**Table 1 nutrients-12-01722-t001:** Characteristics of children and their mothers (*n* = 4059).

Characteristic		Finding	Missing
		*n* (%)	*n* (%)
Social characteristics			
Maternal educational level	Low	420 (10.4)	25 (0.6)
	Mid-low	1215 (30.1)	
	Mid-high	1162 (28.8)	
	High	1237 (30.7)	
Net household income	Low	727 (18.9)	211 (5.2)
	Middle	1036 (26.9)	
	High	2085 (54.2)	
Maternal age at child birth, years, mean (SD)		31.1 (4.8)	0
Children’s characteristics			
Sex	Boy	2057 (50.7)	0
	Girl	2002 (49.3)	
Age, years (SD)		6.0 (0.4)	0
Ethnic background	Western	3040 (74.9)	2 (0.05)
	Non-western	1017 (25.1)	
Weight status	Overweight/obesity	536 (14.5)	373 (9.2)
	Normal weight	3150 (85.5)	

The table is based on a non-imputed dataset.

**Table 2 nutrients-12-01722-t002:** Child lifestyle behaviours by cluster distribution (*n* = 4059).

	Cluster 1 “Relatively Healthy Lifestyle”	Cluster 2 “High Screen Time and Physically Inactive”	Cluster 3 “Physically Active, High Snacks and Sugary Drinks”
	*n* = 1444 (35.6%)	*n* = 1217 (30.0%)	*n* = 1398 (34.4%)
Screen time, mean (SD)	0.99 (0.64)	1.96 (1.10)	1.91 (1.04)
z-score (SE)	−0.59 (0.61)	0.33 (1.05)	0.29 (0.99)
Physical activity, mean (SD)	1.87 (0.96)	0.67 (0.37)	2.26 (1.05)
z-score (SE)	0.21 (0.88)	−0.90 (0.34)	0.56 (0.96)
Calorie-rich snacks, mean (SD)	0.76 (0.60)	1.25 (0.79)	1.95 (0.72)
z-score (SE)	−0.63 (0.64)	−0.11 (0.84)	0.64 (0.77)
Sugary drinks, mean (SD)	1.33 (0.96)	2.48 (1.13)	3.06 (1.16)
z-score (SE)	−0.72 (0.71)	0.13 (0.84)	0.57 (0.86)

The table is based on a non-imputed dataset.

**Table 3 nutrients-12-01722-t003:** Child lifestyle clusters according to child characteristics and socioeconomic status (*n* = 4059).

		Cluster 1 “Relatively Healthy Lifestyle”	Cluster 2 “High Screen Time and Physically Inactive”	Cluster 3 “Physically Active, High Snacks and Sugary Drinks”	*p*-Value *
		*n* (%)	*n* (%)	*n* (%)	
Sex	Boy	676 (46.8)	624 (51.3)	757 (54.1)	<0.001
	Girl	768 (53.2)	593 (48.7)	641 (45.9)	
Weight status	Overweight/obesity	181 (14.0)	178 (15.9)	177 (13.9)	<0.001
	Normal weight	1114 (86.0)	938 (84.1)	1098 (86.1)	
Maternal educational level	Low	73 (5.1)	165 (13.7)	182 (13.7)	<0.001
	Mid-low	329 (22.9)	420 (34.9)	466 (33.5)	
	Mid-high	459 (31.9)	321 (26.6)	382 (27.4)	
	High	576 (40.1)	299 (24.8)	362 (26.0)	
Net household income	Low	178 (13.1)	314 (27.0)	235 (17.7)	<0.001
	Middle	327 (24.0)	323 (27.8)	386 (29.2)	
	High	857 (62.9)	525 (45.2)	703 (53.1)	

The table is based on a non-imputed dataset. * *p*-value is calculated by chi-square test.

**Table 4 nutrients-12-01722-t004:** The association of socioeconomic status with child lifestyle clusters at age 6.

		Cluster 1 “Relatively Healthy Lifestyle”	Cluster 2 “High Screen Time and Physically Inactive”	Cluster 3 “Physically Active, High Snacks and Sugary Drinks”
		OR (95% CI)	OR (95% CI)	OR (95% CI)
Maternal educational level	Low	**0.28 (0.21, 0.37)**	**1.45 (1.13, 1.86)**	**2.28 (1.79, 2.90)**
	Mid-low	**0.46 (0.39, 0.55)**	**1.37 (1.14, 1.64)**	**1.69 (1.42, 2.01)**
	Mid-high	**0.77 (0.66, 0.91)**	1.11 (0.92, 1.34)	1.23 (1.04, 1.47)
	High	Ref	Ref	Ref
Net household income	Low	**0.59 (0.48, 0.74)**	**1.57 (1.27, 1.94)**	1.07 (0.87, 1.31)
	Middle	**0.72 (0.61, 0.84)**	1.18 (0.99, 1.40)	1.22 (1.05, 1.43)
	High	Ref	Ref	Ref

The table is based on an imputed dataset. Models adjusted for child age, gender, ethnic background, and BMI. Bold print indicates statistical significance (*p* = 0.05/6 = 0.008).

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
