# Peer review of "Clustering of Sedentary Behaviours, Physical Activity, and Energy-Dense Food Intake in Six-Year-Old Children: Associations with Family Socioeconomic Status"

_nutrients, 2020, doi:10.3390/nu12061722_

Round 1
Reviewer 1 Report
This is a great research paper that investigates a cluster of lifestyle factors in order to provide a more complete picture of the current situation in the Netherlands. A few points to consider are the following:
1) I understand that they have access to data of children 6 years old only. Thus, it would be nice to explain brief in the introduction and with a few more details in the discussion, what is the age group they belong to and why is important to research these ages.
2) Following the first point, it would be nice to compare these children in the Netherlands with children in other European countries of similar SES and demographic characteristics or not.
3) I also think the authors need to make more sound in the paper, the need for conducting research on lifestyle clusters and what would be the best way to start identifying these clusters.
All in all, great work, pointing research to the right direction!
Reviewer 2 Report
Dear Authors, the submitted manuscript analyses the existing correlations between the family’s socioeconomic status (SES) and some clusters of behaviors of this population of six-year-old children in quite accurately way.
In methodological terms, the study was conducted by the authors in a well-thought and well-designed way, all the variables are clearly defined, and the statistical analysis seems to be appropriate for the type of work. However, despite the value of the analytical part, the discussion part appears to be a little lacking in certain aspects that authors should provide to clarify to make more accessible the social implications of work:
- <>. This evidence maybe represents the principal and central data of this work. It could be useful to spend a few words in explaining the sociological hypothesis underlying these results. Which could be the factor that majorly influences the healthy lifestyle between family economic sources and mother education and why?
- While previous evidences reported that girls were more in the low physical activity clusters, this study showed a higher proportion of boys in the “high screen time and physically inactive” cluster. The authors suggest that these differences may be related to the child’s age. It could be appropriate to clarify in which way the differences in child’s age can justify this trend reversal compared to previous literature data.
Reviewer 3 Report
Dear authors,
Thank you for your interesting work presented to this journal.
The objective of the paper is to examine the lifestyle behaviours patterns as screen time, physical activity, calorie-rich snack consumption and sugar-sweetened beverages consumption, in children aged 6 years in the Netherlands. Secondly. The paper analyses the associations between socioeconomic status (measured by both maternal educational level and net household income) with certain lifestyle behaviours, and the lifestyle behaviours patterns identified.
The conclusions shown the aggregation in three clusters of co-occurring patterns with regard to screen time, physical activity, and energy-dense food intake among children. One third of the children was allocated in the relatively healthy cluster. Other clusters shown healthy or unhealthy trends in co-occurrence of lifestyle behaviours. A higher maternal educational level was associated with a higher odd for the child of the relatively healthy lifestyle behaviour. Children from low income households were more likely to be allocated to one of the relatively unhealthy lifestyle behaviour clusters compared to children from high income household.
The manuscript is written in an English that needs minor improvements to UK English. There are some minor changes to done to improve the paper.
Line 12. …..children aged 6 years ….
Line 82. Change to unify expressions…at child’s age 6 years... to ...when the child was 6 years old.
The study shown is well-designed and well conducted, congratulations. The article is well-organized and contain all the components. The sections are well-developed with a methodology clearly explained. The References section need to improve the format. The article is well-written and easy to understand.
